# Co-Toxicity Factor Analysis Reveals Numerous Plant Essential Oils Are Synergists of Natural Pyrethrins against *Aedes aegypti* Mosquitoes

**DOI:** 10.3390/insects12020154

**Published:** 2021-02-11

**Authors:** Edmund J. Norris, Jeffrey R. Bloomquist

**Affiliations:** 1United States Department of Agriculture, Center for Medical, Agricultural, and Veterinary Entomology, Gainesville, FL 32610, USA; 2Emerging Pathogens Institute, Entomology and Nematology Department, University of Florida, Gainesville, FL 32610, USA; jbquist@epi.ufl.edu

**Keywords:** synergism, antagonism, knockdown, piperonyl butoxide, mosquito, yellow fever mosquito

## Abstract

**Simple Summary:**

With insecticide-resistant mosquito populations becoming an ever-growing concern, new vector control technologies are needed. Plant essential oils represent new insecticides and repellents, which are generally safer to mammals and non-target organisms than conventional materials. A set of 20 plant essential oils was screened alone and in combination with a natural insecticide, pyrethrins, for their ability to produce immobilization (knockdown at 1 h) and mortality at 24 h against the yellow fever mosquito. Overall, only a few of the oils produced considerable mortality or knockdown when applied alone at the doses used in this study. However, a number of them synergized or antagonized the toxicity of natural pyrethrins when applied in mixtures. These findings highlight select plant essential oils that may offer important avenues for the development of future insecticide synergists. Moreover, synergism or antagonism was highly dependent on the amount of the oil applied, e.g., some oils were more effective at improving natural pyrethrins toxicity at low doses, whereas others were better at improving its toxicity at high doses.

**Abstract:**

With insecticide-resistant mosquito populations becoming an ever-growing concern, new vector control technologies are needed. With the lack of new chemical classes of insecticides to control mosquito populations, the development of novel synergists may improve the performance of available insecticides. We screened a set of 20 plant essential oils alone and in combination with natural pyrethrins against *Aedes aegypti* (Orlando) female adult mosquitoes to assess their ability to synergize this natural insecticide. A co-toxicity factor analysis was used to identify whether plant oils modulated the toxicity of natural pyrethrins antagonistically, additively, or synergistically. Both knockdown at 1 h and mortality at 24 h were monitored. A majority of oils increased the toxicity of natural pyrethrins, either via an additive or synergistic profile. Many oils produced synergism at 2 µg/insect, whereas others were synergistic only at the higher dose of 10 µg/insect. Amyris, cardamom, cedarwood, and nutmeg East Indies (E.I.) oils were the most active oils for increasing the mortality of natural pyrethrins at 24 h with co-toxicity factors greater than 50 at either or both doses. A number of oils also synergized the 1 h knockdown of natural pyrethrins. Of these, fir needle oil and cypress oils were the most successful at improving the speed-of-action of natural pyrethrins at both doses, with co-toxicity factors of 130 and 62, respectively. To further assess the co-toxicity factor method, we applied selected plant essential oils with variable doses of natural pyrethrins to calculate synergism ratios. Only the oils that produced synergistic co-toxicity factors produced statistically significant synergism ratios. This analysis demonstrated that the degree of co-toxicity factor correlated well with the degree of synergism ratio observed (Pearson correlation coefficient r = 0.94 at 2 µg/insect; r = 0.64 at 10 µg/insect) and that the co-toxicity factor is a useful tool in screening for synergistic activity.

## 1. Introduction

Plant essential oils are now recognized as effective alternatives to synthetic insecticides against a wide variety of pests. Their generally pleasant aroma, relatively low toxicity to humans and pets, and perceived salubrious qualities have given rise to a large increase in research focused on the development of these agents for pest control [1]. The consumer market has also seen a shift from synthetic insecticides to natural ingredients, as many of these bioactive chemistries are either safer to humans and/or the environment, or perceived to be so [2,3,4]. However, despite the benefits of their use, it is erroneous to assume that all products containing plant essential oils and plant extracts act similarly. To truly appreciate their value, it is necessary to examine their toxic/repellent effects in order to identify the most efficacious chemistries that could be included in future arthropod control technologies [2]. 

Some components in essential oil-based insecticide formulations are considered “generally recognized as safe” (GRAS) by the United States Environmental Protection Agency, under the Federal Insecticide, Fungicide, and Rodenticide Act (FIFRA) [4]. These components are listed in the 25b Exempt category under FIFRA code due to their usage throughout human history, and because they are found in many household food items and fragrances [4,5]. Reclassification of certain components has relaxed the regulatory requirements for registration of plant essential oil-based insecticides and repellents, and allowed for the rapid diversification of novel products. Coupled with the recent consumer-driven push for more natural and “green” control strategies, more of these products are now available for purchase than ever before [3,5]. While not as potent as synthetic insecticides, their potential to kill and repel arthropods is appreciable [6,7,8]. Moreover, many oils have been shown to produce potent synergism with various synthetic insecticides, further demonstrating their utility. Gross et al. [9] demonstrated that various plant essential oils could enhance the toxicity of permethrin against adult female *Aedes aegypti* and *Anopheles gambiae*. Norris et al. [6] showed that the most successful plant oils in the Gross et al. 2015 study could synergize diverse type I and type II pyrethroids, further indicating their activity as synergists. It is important to continually identify novel synergists and toxic additives that could be combined with synthetic insecticides in future insecticidal formulations. 

The goal of this study was to assess the ability of various plant essential oils to improve the toxicity of natural pyrethrins against *Aedes aegypti* of the insecticide-susceptible Orlando strain. For this, we acquired 20 plant essential oils and screened them alone and in combination with natural pyrethrins at discrete doses to characterize their potential to improve knockdown and lethality. To identify and quantify synergism produced by various plant essential oils, we utilized a co-toxicity factor method proposed by Mansour et al. [10]. After calculating the co-toxicity factor for various doses of plant essential oils applied in combination with natural pyrethrins, we calculated traditional synergism ratios for selected oils to evaluate the co-toxicity factor method as a diagnostic tool for the identification of novel synergists.

## 2. Materials and Methods 

### 2.1. Chemicals

Plant essential oils were obtained from Berje, Inc. (Carteret, NJ, USA). Oils were selected for their novelty compared to what has been screened previously in the literature [6,8,9,11,12,13,14]. Natural pyrethrins were obtained from Fairfield American Corp. (Newark, NJ, USA). Pure ethanol (100%) was used as a vehicle for plant oils and natural pyrethrins and was obtained from Sigma Aldrich (St. Louis, MO, USA). Piperonyl butoxide (PBO) (>95%) was also obtained from Sigma Aldrich (St. Louis, MO). Sucrose for sugar water was obtained from Domino (Baltimore, MD, USA). 

### 2.2. Insects

Orlando strain *Aedes aegypti* females were reared according to standard procedures utilized by the United States Department of Agriculture (USDA) [15]. Pupae were provided by the USDA prior to experiments. Pupae were allowed to eclose in 20.3 × 20.3 × 20.3 cm cages (Bioquip, Inc., Rancho Dominguez, CA, USA) and provided 10% sucrose water ad libitum prior to experiments. Mosquitoes were kept in incubators (Darwin Chambers, St. Louis, MO, USA) maintained at 28 °C ± 75% humidity with a 12:12 light dark cycle. Only non-blood-fed adult females were used for each experiment, and treated individuals ranged from 2–7 days old [16].

### 2.3. Topical Application

Topical applications of ethanolic solutions containing natural pyrethrins and/or essential oils were performed using methods similar to those outlined in [6]. Concentrations of natural pyrethrins ranged from those producing 10% to 90% mortality to ensure a good fit for probit models and to accurately calculate LD_25_ and LD_50_ values. Plant essential oils were applied at either 2 µg/insect or 10 µg/insect alone or in combination with the approximate LD_25_ of natural pyrethrins. The LD_25_ (1.6 ng/mosquito) was calculated using a Probit model and given as a discrete dose assuming the weight of female mosquitoes was 2.85 ± 0.08 mg/mosquito (average from five cohorts with SEM). Topical applications were performed by anesthetizing adult female mosquitoes on ice for 5 min, and then placing them on a cold, glass Petri dish to prevent reanimation. A Whatman No. 2 filter paper was used to prevent mosquitoes from coming in contact with excess condensation. A 0.2 µL volume of each treatment was applied to the pronotum of each mosquito using a Hamilton (Reno, NV, USA) repeating applicator with a gastight Hamilton syringe and placed in 470 mL deli paper cups. At least five concentrations were used to obtain LD_25_ and LD_50_ values for natural pyrethrins alone or in combination with select oils. Ten mosquitoes were treated for each concentration representing one replicate, and at least three different cohorts of mosquitoes were used for each concentration in an effort to control for biological variability in the mosquitoes produced by the Orlando colony. A cohort was defined as a distinct rearing group (i.e., adult mosquitoes obtained from distinct batches of eggs placed in water to initiate development each week). Controls were performed using the vehicle, ethanol. Tulle fabric, fastened to the cup with a deli cup lid rim (center removed), was used to keep mosquitoes from escaping. Deli cups containing ten treated mosquitoes per concentration per replicate were placed in an incubator for the remainder of the experimental interval and kept at the same temperature and light cycle as during rearing. Knockdown, defined as the inability to fly or maintain normal standing posture, was recorded at 1 h, and mortality was recorded at 24 h. Mortality was defined as no movement, even after gently tapping the cup several times to assess response. 

### 2.4. Data Analysis

To assess synergism, the co-toxicity factor method was utilized [10]. In short, percentage knockdown and mortality was recorded for all treatments of plant essential oils alone, natural pyrethrins alone, and combinations thereof, and co-toxicity factors were calculated using the following equation: 


$$Co-toxicity\;Factor=\;\frac{Observed\;Mortality-Expected\;Mortality}{Expected\;Mortality}\times 100$$


In this equation, observed mortality was the toxicity observed experimentally in combinations of plant oils and natural pyrethrins at either specific dose of plant essential oils (2 and 10 µg/insect). Expected mortality was the additive sum of the observed mortality for natural pyrethrins alone and each plant essential oil alone. Values > 20 represent synergistic mixtures, −20 ≤ values ≤ 20 represent additive mixtures and values < −20 represent mixtures that are antagonistic. An LD_25_ of natural pyrethrins was used alone or in combination with discrete doses of plant essential oils to calculate co-toxicity values. LD_50_ and LD_25_ values for natural pyrethrins were derived from Probit analysis (Finney et al. 1952 [17]), using a PROC Probit calculation with a control correction option (OPTC) (to account for control mortality) performed in SAS 9.4 (SAS Institute, Inc., Cary, NC, USA). After synergistic combinations were identified using the co-toxicity factor method, conventional synergism ratios (LD_50_ of natural pyrethrins alone ÷ LD_50_ of natural pyrethrins + a sublethal dose of essential oil) were calculated to evaluate the degree of synergism produced. These experiments were done for comparison and to further evaluate the performance of the co-toxicity factor method outlined by Mansour et al. 1966 [10]. To compare effects at discrete doses to one another or between those and control, an ANOVA (α = 0.05) with a Tukey’s post-test was used with an α value of 0.05. For synergism assessments at discrete doses of plant essential oils, natural pyrethrins were applied in every cohort to obtain a relevant percentage mortality for the LD_25_ of natural pyrethrins for each cohort. As the response to natural pyrethrins differed slightly in every cohort, this approach ensured that cohort bias was accounted for in all comparisons in the analysis. As a result, the responses for natural pyrethrins alone and plant essential oils alone were adequately taken into account in the co-toxicity calculations for each combination of oil and natural pyrethrins using the aforementioned equation.

## 3. Results

Overall, plant essential oils exhibited a spectrum of activity when applied both alone and in combination with natural pyrethrins. Unique differences were observed in knockdown at 1 h and mortality at 24 h for each essential oil. At 2 (Table 1) and 10 µg/insect (Table 2), a majority of oils did not produce any effects significantly different from the control when applied alone. Only balsam (Peru) produced significant knockdown at 2 µg/insect compared to the control (Table 1), but the overall mean level of knockdown was low; 20 ± 5.8% at 1 h (mean + SEM). Moreover, at the 2 µg/insect dose, no oils produced significant mortality compared to the control, but at 10 µg/insect (Table 2), more plant essential oils caused significant knockdown and mortality compared to the ethanol control. Piperonyl butoxide (PBO), Amyris, balsam (Peru), cypress, and guaiacwood all produced significant 1 h knockdown, and among these, balsam (Peru) (73.3 ± 12%) produced the highest levels of 1 h knockdown at this screening concentration. Piperonyl butoxide (PBO), Amyris, Canadian balsam fir, citronella, and guaiacwood all produced statistically significant mortality compared to the ethanol control at 24 h when applied at 10 µg/insect; however, only PBO, Canadian balsam fir, and guaiacwood produced mortality that was greater than 50% (Table 2).

In order to screen for synergistic interactions between natural pyrethrins and plant essential oils, they were applied with a calculated LD_25_ of natural pyrethrins. The theoretical LD_25_ and LD_50_ for natural pyrethrins were calculated to be 0.55 ng/mg insect and 1.53 ng/mg insect, respectively. As the weight for each mosquito cohort varied, we applied a discrete dose (1.6 ng/insect) assuming mosquitoes in each cohort weighed 2.85 mg/mosquito (average of five cohorts of mosquitoes). The theoretical LD_25_ produced 14 ± 1.9% mortality at 24 h when averaged across all replicates. The range of average mortality for all cohorts (groups of three or more replicates associated with selected plant essential oils within each group) was 6.6–22% mortality. As mortality produced by the theoretical dose was lower than expected (i.e., below 25%), we calculated the co-toxicity factors using the actual percentage mortality produced by natural pyrethrins within each cohort to avoid cohort biases.

Of the various oils, some produced synergism when applied with natural pyrethrins at 2 µg/insect, both of knockdown and mortality (Table 1). Cedarwood (Virginian), dillseed, fir needle oil, fennel, and parsley all produced knockdown co-toxicity factors greater than 20. In this assessment, fir needle oil performed the best with a knockdown co-toxicity factor of 77.6. This response was largely driven by the lack of knockdown produced by fir needle oil and the relatively high increase in knockdown for the mixture (53.3 ± 18.6%). Many other oils increased the knockdown effects of natural pyrethrins, but these were essentially additive, producing co-toxicity factors between −20 and 20 (Table 1). A number of plant essential oils also antagonized knockdown at 1 h. PBO antagonized knockdown with only 10 ± 3.7% knockdown observed in combinations of PBO and natural pyrethrins, whereas 47% knockdown was observed for natural pyrethrins alone. Of the oils, balsam (Copaiba), cade, Canadian balsam fir, dillweed, and ginger root all antagonized the quick immobilizing character of natural pyrethrins at 1 h, with co-toxicity factors less than −20. Of these, Canadian balsam fir was the most antagonistic (co-toxicity factor = −63.6). At the screening concentration of 2 µg/insect, numerous oils/agents synergized mortality. PBO, the commercial synergist standard used in this study, provided a high degree of synergism with a co-toxicity factor of 71 (Table 1). Amyris, balsam (Copaiba), cedarwood (Texas), cedarwood (Virginian), dillseed, fennel, fir needle oil, and nutmeg E.I. all produced synergism with co-toxicity factors greater than 20. Of these, nutmeg E.I. produced the largest co-toxicity factor (300). The other oils produced additive increases in mortality at 24 h, with the exception of cade, cardamom, cypress, dillweed, galbanum, and ginger root, which produced antagonistic co-toxicity factors (Table 1). 

At the 10 µg/insect screening concentration, a number of oils synergized the 1 h knockdown and 24 h mortality produced by natural pyrethrins (Table 2). Cade, cardamom, cedarwood (Virginian), cypress, dillweed, fir needle, and guaiacwood oils synergized the knockdown of natural pyrethrins at 1 h, with fir needle oil being the most successful. Other oils either additively increased or antagonized 1 h knockdown. PBO strongly antagonized 1 h knockdown by natural pyrethrins at the 10 µg/insect concentration with a co-toxicity factor of −115, which was the most negative co-toxicity factor observed in this study. Among the oils, Amyris, balsam (Copaiba), balsam (Peru), Canadian balsam fir, cedarwood (Texas), and galbanum antagonized natural pyrethrins 1 h knockdown, whereas cedarleaf, dillseed, fennel, ginger root, nutmeg E.I., and parsley all additively increased 1 h knockdown of natural pyrethrins (Table 2). Many oils synergized natural pyrethrins mortality at 24 h at the 10 µg/insect concentration, whereas PBO did not. PBO instead produced high mortality when applied alone and the improvement of the combined mixture was minimal. Among the synergistic oils, Amyris, balsam (Copaiba), cardamom, cedarleaf, cedarwood (Texas), cedarwood (Virginian), dillseed, dillweed, fennel, ginger root, nutmeg E.I., and parsley, it was found that cardamom was the most active with a co-toxicity factor of 200. The remaining oils, with the exception of Canadian balsam fir and galbanum, increased the 24 h mortality of natural pyrethrins by an additive extent. Both Canadian balsam fir and galbanum antagonized the toxicity of natural pyrethrins at this concentration. Overall, the results in Table 1 and Table 2 show that the synergistic potential of plant essential oils and PBO is concentration dependent. 

To assess the utility and accuracy of the co-toxicity factor metric, we screened a number of plant essential oils and PBO (applied at a sublethal dose of 2 µg/insect) with variable concentrations of natural pyrethrins. The LD_50_ values for natural pyrethrins in these experiments were then compared to the original LD_50_ of natural pyrethrins to obtain a synergism ratio (Table 3). Of the oils screened in these follow-up studies, Amyris oil + natural pyrethrins produced the lowest LD_50_ value (0.21 ng/mg mosquito) and the highest synergism ratio (7.3). Cedarwood (Texas) also produced significant synergism, but the synergism ratio for this formulation was 4, very similar to that produced by PBO (5.1). The other cedarwood oil, Virginian, also produced significant synergism with a synergism ratio of 3.6. Interestingly, both fir needle oil and citronella oil produced low levels of toxicity synergism, but these were not significant as indicated by *t*-test; *p* = 0.59 for fir needle oil and *p* = 0.74 for citronella. Slope values for natural pyrethrins were relatively unchanged by combined application with these plant essential oils.

## 4. Discussion

Twenty plant essential oils were screened in combination with natural pyrethrins and knockdown at 1 h and mortality at 24 h were determined. Many of the oils used in this study have not been screened previously in combination with natural pyrethrins and/or have not been assessed as synergists on *Aedes aegypti* female mosquitoes. Moreover, assessing enhancement of both knockdown and mortality are important considerations in the characterization of novel public health pest control formulations, as both effects may lead to the prevention of host feeding. Norris et al. [12] proposed that knockdown of intoxicated mosquitoes may lead to higher levels of mortality over time in the field, due to exposure to fungal pathogens, desiccation or starvation through the inability to feed, and increased predation. The present study demonstrated that many plant essential oils not only improve the mortality produced by natural pyrethrins at 24 h, but also improve their speed-of-action. 

In order to characterize the synergistic potential of plant essential oils, we first screened them alone to better understand their toxicological contributions in our mixtures. A wide range of toxicities were observed, with balsam (Peru) producing the most significant knockdown at 1 h at the 10 µg/insect concentration. Seo et al. [18] demonstrated that balsam (Peru) is predominantly composed of benzyl benzoate and benzyl cinnamate. These constituents may be useful leads for future insect control formulations, either as natural insecticides or agents that improve the knockdown effects of currently available insecticide formulations. In addition, a significant amount of recovery was observed in mosquitoes treated with this oil (73.3% knockdown at 1 h and 26.7% mortality at 24 h), indicating that metabolic processes probably detoxified the constituents within balsam (Peru) oil. Therefore, additional synergists such as PBO or S,S,S-tributylphosphorotrithioate (DEF) should increase the 24 h toxicity of this oil [19]. Guaiacwood oil and Canadian balsam fir were the most toxic at 24 h, indicating their potential as natural insecticides. Norris et al. [6] demonstrated that guaiacwood oil was predominantly composed of guaiol along with the minor constituent sesquiterpenoids, bulnesol, and bulnesene. Canadian balsam fir is predominantly composed of α-pinene, β-pinene, and phellandrene, any of which may be responsible for its toxicity [20]. Further work is needed to characterize the biological activity of these oils and assess the contribution of each individual constituent towards overall toxicity. 

Formulation additives can significantly augment the speed-of-action of select insecticides; therefore, understanding their contribution to speed-of-action is an important consideration, whether additive or synergistic [21]. Five oils synergized 1 hr knockdown at the 2 µg dose and six the knockdown observed at the 10 µg dose. These findings warrant future exploration as there may be agents within these oils that significantly potentiate the effects of natural pyrethrins directly on the insect nervous system or facilitate the penetration of natural pyrethrins across the cuticle. Screening the individual constituents from these oils directly on the nervous system in combination with pyrethroids will identify constituents acting via this mechanism. Moreover, if these oils increase insecticide penetration, it would be valuable to better understand the physicochemical factors underlying these activities. Increased penetration may be facilitated via improved passive diffusion across the cuticle (as in the case of calcofluor [22,23]) or through the inhibition of drug efflux pumps, similar to the mechanism of verapamil, an ATP-binding cassette (ABC) transporter inhibitor [24]. 

In contrast to the generally positive effects of essential oils on knockdown by natural pyrethrins, PBO significantly reduced knockdown by this natural insecticide. Norris et al. [10] showed previously that PBO at both 2 and 10 µg/mosquito could significantly decrease the knockdown produced by select pyrethroids at 1 h post-application, and Kasai et al. [17] showed that 1 hr pretreatment with PBO significantly decreased the penetration rate of [^14^C]-permethrin into *Aedes aegypti* females. These findings advise against combining diverse formulation additives simply due to their independent activity. Our study corroborated the earlier findings that PBO significantly slowed the immobilization produced by natural pyrethrins in both the 2 µg and 10 µg/insect applications. 

Synergism of lethality was observed using the co-toxicity method, identifying oils that were synergistic with natural pyrethrins at the low dose application level (2 µg/insect) but not synergistic at the high dose level (10 µg/insect), oils that were synergistic at only the high dose level, those synergistic at both doses, and some that were not synergistic at either dose. This variety of responses was also true for antagonism. These findings demonstrate that synergist concentration must be an important consideration in the development of novel insecticidal mixtures. Unexpectedly, some oils and PBO were synergistic only at the low dose, but not at the high dose, which may result from sequestration of pyrethrins at the cuticular boundary. Moreover, it is possible that at the high dose, excess oil/ PBO did not fully penetrate the insect cuticle. Of these oils, nutmeg E.I. produced the largest co-toxicity factor (Table 1), perhaps due to the constituents safrole and myristicin, natural compounds that are structurally similar to PBO [25] and contain an identical methylenedioxyphenyl moiety [26]. Yang et al. 2015 demonstrated that myristicin was capable of inactivating human CYP1A2 (cytochrome P450 monooxygenase) via mechanism-based inhibition [27], similar to the action of PBO [26,27]. At 10 µg/insect, cardamom produced the highest co-toxicity factor (200) seen at this dose. This oil is predominantly composed of complex mixtures of oxygenated monoterpenoids [28]. Further work will be needed to separate these constituents to ascertain their specific bioactivity and mechanisms of action. While this study identified a number of novel synergistic plant essential oils, some apparent differences from other studies were noted. For example, Tak et al. [8] showed that 10 µg/female doses of cedarwood, dill, and fennel oils applied in combination with permethrin did not produce statistically significant mortality greater than permethrin alone, whereas cedarwoods, dillweed and dillseed, and fennel oils synergized natural pyrethrins at the specific doses studied here. The greater potential of plant essential oils to increase the toxicity of natural pyrethrins vs. permethrin may result from greater susceptibility of natural pyrethrins to metabolic degradation than permethrin. If this is the case, inhibition of metabolic processes by plant essential oils may more significantly increase the toxicity of natural pyrethrins than permethrin. Norris et al. put forth a similar justification after finding type I pyrethroids were more strongly synergized by plant essential oils than type II pyrethroids [12]. 

To further evaluate the co-toxicity factor method, we selected a number of plant essential oils that produced co-toxicity factors greater than 20 (and those that produced co-toxicity factors between −20 and 20) to assess whether this method translated well to the established LD_50_ ratio method of identifying synergism. If the co-toxicity factor method was meaningful and scientifically sound, it should translate well to another metric commonly utilized to measure synergism in the literature [16,29,30]. In these studies, oils that produced co-toxicity factors greater than 20 also produced significant synergism ratios, such as Amyris, cedarwood (Virginian), and cedarwood (Texas). Moreover, the two oils that did not produce additive co-toxicity factors (i.e., between −20 and 20), citronella and fir needle oil, did not produce statistically significant synergism ratios. Correlation between traditional synergism ratios and co-toxicity factors at the 2 µg/insect level was quite strong (Pearson Correlation r = 0.94) (Figure 1). For example, Amyris produced the largest synergism ratio and the largest co-toxicity factor. Direct correlation was less pronounced when comparing the synergism ratios obtained using 2 µg/insect + NP with the co-toxicity factors obtained using 10 µg/insect of oil (Pearson Correlation r = 0.64). It is possible that the slope values of combined mixtures may directly affect the degree of co-toxicity factors observed at low or high potency screening concentrations, in addition to any number of toxicokinetic/dynamic factors. Differences in the ability of plant essential oils to inhibit metabolic processes or aid/hinder penetration of natural pyrethrins may also differ at each respective dose applied (2 or 10 µg). More work needs to be performed to comprehensively evaluate the strengths and weaknesses of this method, but our study supports the throughput and merit of co-toxicity metric analysis.

Insecticide synergists improve the efficacy of various synthetic and natural insecticides, potentially allowing them to overcome insecticide resistance in the field [31]. Future work should be done to assess how these affect insecticide resistance to various insecticidal classes. Previous studies have shown that specific plant essential oils increase the toxicity of permethrin and deltamethrin on both pyrethroid-resistant and pyrethroid-susceptible strains of *Ae. aegypti* and *Anopheles gambiae* [13,14]. As many of the essential oils screened in this exploration were not screened against insecticide-resistant strains, it is imperative to elucidate their potential as resistance-breaking insecticide additives. Moreover, Kumar et al. [32] demonstrated that mosquito strains selected with deltamethrin over 40 generations were 60% less resistant when selected against combinations of PBO and deltamethrin (compared to deltamethrin alone). This suggests synergists may not only play a role after the development of insecticide resistance, but may serve to slow its development, also. 

## 5. Conclusions

This study demonstrates the potential of a set of plant essential oils to selectively enhance or antagonize natural pyrethrins when both are applied in combination. Not only did some of these oils synergize natural pyrethrins-based mortality at 24 h, they also increased its ability to immobilize insects shortly after exposure (knockdown at 1 h). However, not all oils produced synergism, with many producing antagonisms of natural pyrethrins toxicity. In fact, synergism and/or antagonism was highly dose dependent. This study demonstrates the utility of select plant essential oils as leads for the development of future insecticide synergists. Many of the oils identified as synergists of either knockdown or lethality have not been studied before as synergists against *Ae. aegypti* mosquitoes. Further, the dose dependence of synergistic/antagonistic interactions demonstrates that the dose of individual agents in insecticidal mixtures should be carefully considered. 

## Figures and Tables

**Figure 1 insects-12-00154-f001:**
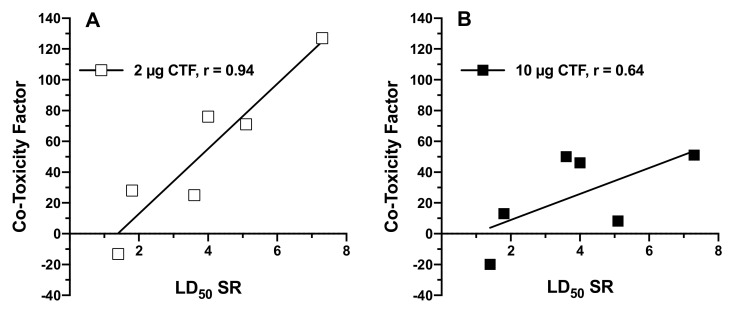
Synergism ratios of LD_50_ values (SR) plotted against co-toxicity factors (CTF) obtained at (**A**) 2 µg/insect and (**B**) 10 µg/insect, along with linear regression analysis. Strong linear correlation value was observed for the 2 µg/insect (Pearson’s correlation coefficient r = 0.94), whereas at 10 µg/insect moderate linear correlation was observed (Pearson’s correlation coefficient r = 0.64).

**Table 1 insects-12-00154-t001:** Percentage 1 h knockdown and 24 h mortality values for natural pyrethrins (LD_25_), plant essential oils or PBO synergists (2 µg/insect), mixture of natural pyrethrins + synergist, and the calculated co-toxicity factors for each mixture.

Essential Oil/Compound	1 h % Knockdown ± SE				24 h % Mortality ± SE			
	Natural Pyrethrins	Synergist	Mixture	Co-toxicity Factor	Natural Pyrethrins	Synergist	Mixture	Co-Toxicity Factor
Control (ethanol)	NA	0.6 ± 0.6	NA	NA	NA	1.8 ± 1	NA	NA
PBO	46.7 ± 3.7	6.7 ± 3.7	10 ± 3.7	−81.2	8.9 ± 3.3	6.7 ± 2.4	26.7 ± 3.7	**71.2**
Amyris	48.5 ± 5.1	6.7 ± 3.3	63.3 ± 21.9	14.7	11.4 ± 2.6	3.3 ± 3.3	33.3 ± 8.8	**127**
Balsam Copaiba	50 ± 5.8	2.5 ± 2.5	32.5 ± 4.8	−38.1	13.8 ± 3.7	2.5 ± 2.5	25 ± 2.9	**53.4**
Balsam Peru	58 ± 8	20 ± 5.8 *	60 ± 10	−23	18 ± 5.8	10 ± 10	26.7 ± 12	−4.6
Cade	23.3 ± 6.6	6.6 ± 6.6	23.3 ± 18.6	−22.1	6.6 ± 3.3	10 ± 5.8	3.3 ± 3.3	−80.1
Canadian Balsam Fir	40 ± 5.8	15 ± 8.8	20 ± 0	−63.6	13.3 ± 8.8	20 ± 0	23.3 ± 3.3	−21
Cardamom	60 ± 5.8	0 ± 0	60 ± 5.8	0	20 ± 10	0 ± 0	10 ± 10	−50
Cedarleaf	58 ± 8	3.3 ± 3.3	56.7 ± 3.3	−7.5	18 ± 5.8	13.3 ± 8.8	36.7 ± 8.8	17.3
Cedarwood Texas	47.8 ± 4.9	4 ± 2.4	44 ± 9.3	−15.1	14.4 ± 5.3	14 ± 7.5	50 ± 8.9	**76.1**
Cedarwood Virginian	36 ± 4	0 ± 0	63.3 ± 8.8	**75.8**	18 ± 9.2	6 ± 4	30 ± 7.1	**25**
Citronella	29 ± 3	1.3 ± 1.3	18.6 ± 2.7	−39.2	29.3 ± 2.8	4 ± 4	29.3 ±3.5	−13.1
Cypress	50 ± 4.5	5 ± 5	62.5 ± 15.5	13.6	13.3 ± 2.1	5 ± 5	12.5 ± 6.3	−32.4
Dillseed	26.7 ± 3.3	0 ± 0	40 ± 17.3	**49.8**	6.7 ± 3.3	3.3 ± 3.3	16.6 ± 16.6	**66**
Dillweed	60 ± 5.8	6.7 ± 3.3	53.3 ± 6.7	−20	20 ± 10	10 ± 5.8	16.7 ± 3.3	−44.3
Fennel	50 ± 5.8	2.5 ± 2.5	70 ± 9.1	**33.3**	13.8 ± 3.7	5 ± 5	25 ± 15	**33**
Fir Needle Oil	30 ± 5.8	0 ± 0	53.3 ± 18.6	**77.6**	20 ± 10	3.3 ± 3.3	30 ± 11.5	**28.8**
Galbanum	40 ± 4.5	2.5 ± 2.5	47.5 ± 11.1	11.8	12 ± 7.3	6 ± 6	4 ± 4	−77.8
Ginger Root	40 ± 5.8	0 ± 0	30 ± 10	−25	13.3 ± 8.8	13.3 ± 8.8	13.3 ± 6.6	−50
Guaiacwood	30 ± 0	10 ± 5.8	36.7 ± 13.3	−8.25	6.6 ± 3.3	10 ± 5.8	16.7 ± 6.7	0
Nutmeg EI	47.1 ± 5.2	3.3 ± 3.3	53.3 ± 6.6	5.8	10 ± 3.1	0 ± 0	40 ± 20	**300**
Parsley	60 ± 5.8	6.7 ± 6.7	83.3 ± 6.6	**24.9**	20 ± 10	13.3 ± 3.3	26.7 ± 21.9	−19.8

denotes statistically significant percentage from the control via ANOVA (α = 0.05) with a Tukey’s post-test; Bold numerals represent synergistic combinations; Red numerals represent antagonistic combinations.

**Table 2 insects-12-00154-t002:** Percentage 1 h knockdown and 24 h mortality values for natural pyrethrins (LD_25_), plant essential oils or piperonyl butoxide (PBO) synergists (10 µg/insect), mixture of natural pyrethrins + synergist, and the calculated co-toxicity factors for each mixture.

Essential Oil/Compound	1 h % Knockdown ± SE				24 h % Mortality ± SE			
	Natural Pyrethrins	Synergist	Mixture	Co-Toxicity Factor	Natural Pyrethrins	Synergist	Mixture	Co-Toxicity Factor
Control (ethanol)	NA	0.6 ± 0.6	NA	NA	NA	1.8 ± 1	NA	NA
PBO	46.7 ± 3.7	21.7 ± 4 *	21.7 ± 4.5	−115	8.9 ± 3.3	65 ± 8 *	80 ± 8.3	8.2
Amyris	48.5 ± 5.1	47.5 ± 13.8 *	57.5 ± 13.1	−40	11.4 ± 2.6	35 ± 16.6 *	70 ± 12.2	**50.8**
Balsam Copaiba	50 ± 5.8	5 ± 2.9	40 ± 4.1	−27.3	13.8 ± 3.7	10 ± 5.8	30 ± 26.3	**26.1**
Balsam Peru	58 ± 8	73.3 ± 12 *	76.7 ± 14.5	−41.6	18 ± 5.8	26.7 ± 8.8	43.3 ± 8.8	−9.2
Cade	23.3 ± 6.6	23.3 ± 13.3	56.7 ± 23.3	**21.7**	6.6 ± 3.3	26.7 ± 12	30 ± 11.5	−9.9
Canadian Balsam Fir	40 ± 5.8	25 ± 5	30 ± 10	−53.8	13.3 ± 8.8	70 ± 11.5 *	60 ± 10	−28
Cardamom	60 ± 5.8	0 ± 0	83.3 ± 8.8	**38.8**	20 ± 10	0 ± 0	40 ± 17.3	**200**
Cedarleaf	58 ± 8	6.7 ± 3.3	73.3 ± 6.6	13.3	18 ± 5.8	6.7 ± 3.3	50 ± 11.5	**102**
Cedarwood Texas	47.8 ± 4.9	8.3 ± 3.1	40 ± 6.8	−28.7	14.4 ± 5.3	23.3 ± 10.5	55 ± 14.3	**45.9**
Cedarwood Virginian	36 ± 4	6.7 ± 3.3	66.7 ± 8.8	**56.2**	18 ± 9.2	18 ± 8.6	54 ± 12.9	**50**
Citronella	29 ± 3	4 ± 0	20 ± 4.6	−39.4	22 ± 7.2	37.3 ± 6.1 *	47.5 ± 6.3	−19.9
Cypress	50 ± 4.5	10 ± 0 *	97.5 ± 2.5	**62.5**	13.3 ± 2.1	5 ± 2.9	22.5 ± 6.3	**21.6**
Dillseed	26.7 ± 3.3	23.3 ± 3.3	43.3 ± 18.6	−13.4	6.7 ± 3.3	13.3 ± 6.6	26.7 ± 8.8	**33.5**
Dillweed	60 ± 5.8	0 ± 0	76.7 ± 8.8	**27.8**	20 ± 10	0 ± 0	33.3 ± 6.7	**66.5**
Fennel	50 ± 5.8	10 ± 4.1	52.5 ± 2.5	−12.5	13.8 ± 3.7	7.5 ± 2.5	30 ± 9.1	**40.8**
Fir Needle Oil	30 ± 5.8	3.3 ± 3.3	76.6 ± 8.8	**130**	20 ± 10	6.6 ± 6.6	30 ± 5.8	12.7
Galbanum	40 ± 4.5	7.5 ± 7.5	35 ± 14.4	−26.3	12 ± 7.3	12 ± 12	10 ± 5.5	−58.3
Ginger Root	40 ± 5.8	10 ± 10	45 ± 5	−10	13.3 ± 8.8	6.6 ± 3.3	33.3 ± 3.3	**67.3**
Guaiacwood	30 ± 0	43.3 ± 13.3 *	56.7 ± 12	**22.6**	6.6 ± 3.3	66.7 ± 8.8 *	66.7 ± 8.8	−9
Nutmeg EI	47.1 ± 5.2	7.5 ± 4.8	57.5 ± 6.3	5.3	10 ± 3.1	2.5 ± 2.5	27.5 ± 12.5	**120**
Parsley	60 ± 5.8	23.3 ± 12	83.3 ± 8.8	0	20 ± 10	3.3 ± 3.3	46.7 ± 12	**100**

* denotes statistically significant percentage from the control via ANOVA (α = 0.05) with a Tukey’s post-test; Bold numerals represent synergistic combinations; Red numerals represent antagonistic combinations.

**Table 3 insects-12-00154-t003:** Dose–response statistics for natural pyrethrins applied alone and in combination with select candidate synergists and their respective synergist ratios.

Treatment	N	LD_50_ ng/mg Insect (95% CI)	Slope (SE)	Synergism Ratio *
Natural pyrethrins (NP)	290	1.53 (1.0–3.3)	1.5 (0.3)	-
NP + PBO	210	0.3 (0.19–0.44)	2.0 (0.53)	5.1 *
NP + citronella	150	1.13 (0.7–5.2)	1.7 (0.49)	1.4
NP + Amyris	150	0.21 (0.13–0.3)	2.0 (0.3)	7.3 *
NP + Cedarwood (Virginian)	150	0.43 (0.25–0.77)	1.6 (0.49)	3.6 *
NP + Cedarwood (Texas)	150	0.38 (0.26–0.57)	1.9 (0.36)	4 *
NP + fir needle oil	200	0.85 (0.6–1.2)	2.2 (0.38)	1.8

* denotes statistically significant percentage from the control (NP alone) via lack of overlap in 95% confidence intervals.

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
