# Peer review of "Co-Toxicity Factor Analysis Reveals Numerous Plant Essential Oils Are Synergists of Natural Pyrethrins against Aedes aegypti Mosquitoes"

_insects, 2021, doi:10.3390/insects12020154_

Round 1

Reviewer 1 Report

The authors have more or less adequately addressed my previous concerns. I have no additional comments and recommend this manuscript for publication. 

Author Response

We thank the reviewer for their comments and their aid in improving the manuscript for final publication. 

Reviewer 2 Report

Dear authors,

Thank you for considering my comments on the first version of your manuscript so that I propose to be published.

Author Response

We thank the reviewer again for their comments

Reviewer 3 Report

Authors have responded to the reviewers' comments appropriately.

Author Response

We thank the reviewer again for their comments. 

Reviewer 4 Report

The present study addresses the issue of synergism between pyrethins and essential oils as adulticides of Aedes aegypti. This falls within the scope of the journal Insects.
The main issue I have with the study is the lack of a statistical basis for the co-toxicity factor analyses. There are several alternative approaches that have a sound statistical basis. This needs to be made clear at several points by the authors. In particular, I have read Mansour et al. 1966 and the source from which they obtained the concept of co-toxicity factor (Sun & Johnson 1960). Mansour et al. do not justify their classification of the >-20; -20 - +20; >20 intervals as anything other than arbitrary. Sun & Johnson do not mention these classifications.
This is a key point as Norris & Bloomquist describe their co-toxicity factor results as "significant" in many places in which no statistical test was applied and no probability calculated. As three replicates were performed (replication in the present study was low), I suggest that co-toxicity factors could be subjected to ANOVA or non-parametric analysis to identify statistically valid differences. This needs to change throughout the Results and Discussion.
Other points
L17, remove point after mosquito
L20, delete "we found that"
L33, should read: were monitored
L97, provide details or a published source on the mosquito rearing procedures
L126, should read: at 24 h (delete hr.)
L132, suggest you insert the text: ....factors were calculated as percentages using the.... to make it clear that these are percentage values.
L142, Finney reference needs to be numbered.
L143, delete correction
L148, Mansour reference needs to be numbered. Are you citing Mansour et al. 1962 or 1966? Ref [10] is 1966.
L160, delete specific
L162, Balsam (Peru) not labeled with asterisk in Table 1, but the other Balsam is wrongly labeled.
Table 1, Table 2 - edit label in Co-toxicity factor columns to "Co-toxicity factor (%)". This will make it clear that a particular value is a percentage not a multiple, e.g. 80 is 80%, not an 80-fold difference.
L166, what about cypress oil in Table 2?
L234, delete: degree of (there are no degrees of mortality). Should read: ...produced high mortality...
L241, change data to "results"
L247, 0.15 is not the LD value that appears in Table 3.
L269, mosquitoes in the field feed on nectar and perhaps honeydew (only sugar-water in the laboratory).
L276, should read: Seo et al. [16]....
L287, change drive to "be responsible for"
L298, change driving to "underlying"
L355-356, the reason why co-toxicity factor was used rather than more statistically rigorous modern approaches (e.g. doi: 10.1007/978-94-017-9795-5_4, or doi: 10.1002/etc.3878, but there are also others) is difficult to justify in my opinion.
L361, L376 - change Aedes to Ae.
L365, should read generations
L367, should read: ...may also serve....
L370, should read: oils
L371, change to: ....pyrethrin-induced mortality...

Author Response

We have addressed the comments in the attached word file. 

Round 2

Reviewer 4 Report

The authors have addressed the majority of my concerns. I accept their argument that the co-toxicity factor is a descriptive approach to identifying potential synergistic/antagonistic interactions.
The authors themselves state "Statistical significance is not a feature of the [co-toxicity] metric." However, the word "significant" occurs 40 times in the manuscript and gives the impression of statistical significance in the use of the metric.
I would ask the authors to remove the term significant/significantly in the entire manuscript where it is related to the co-toxicity factor metric. This term should only be retained where it describes probabilities derived from the Probit/ANOVA/Tukey statistical procedures.
I believe that the manuscript would then be suitable for publication.

Author Response

We have addressed the reviewer's request to remove the term "significant" when used to describe co-toxicity factors. We have left the term when it is used in relation to ANOVA, Probit, or other statistical tests. 

This manuscript is a resubmission of an earlier submission. The following is a list of the peer review reports and author responses from that submission.

Round 1

Reviewer 1 Report

I think overall this is a conceptually sound study that is of some importance for expanding our knowledge of potential chemical products that may improve mosquito control and disease mitigation. However, as it is currently written, I have some concerns about its publication, and therefore must recommend a major revision.

The most important thing to address is the absence of any mention (so far as I can tell) of sample sizes. These are included in Table 3, but not in the other tables nor in the text. Without the reporting of sample sizes, it is impossible to assess how much importance should be afforded the results observed. Furthermore, you mention ‘cohorts’ but make no attempt to explain how these were defined, how many there were, how many individuals per cohort, etc. This absolutely needs to be rectified if this manuscript is to be published.

From the methods: “Natural pyrethrins were applied in every cohort to obtain a relevant percentage mortality for the LD25 of natural pyrethrins for each cohort. This approach ensured that cohort bias was accounted for in all comparisons in the analysis.” This is a wholly inadequate description of these methods and makes it impossible for your readers to evaluate or interpret the significance of you results. Likewise, it would be impossible for anyone to replicate your experiments based on your description of the methods used (this is just one example of the lack of detail; there are many others!)

A second concern is that the results are inconsistently presented, vague and hard to interpret as currently presented. I think the authors need to strongly consider altering or expanding their presentation of findings to make them more easily interpretable to readers. What I think would be most useful would be box and whisker plots showing the median/mean & variation of % knockdown and % mortality, with straight pyrethrin included so that visually the reader can very clearly see what oils (alone and in combination) reduced or increased these metrics. Horizontal lines could then demarcate the antagonistic, additive, and synergistic interactions. For data like these, I prefer box & whisker plots to something like a bar graph, but even bar graphs (as in your 2019 paper) would be an improvement. Also, presenting the data for the oils along, or when combined with pyrethrin I think makes more sense.

I think a better job needs to be done explaining what is meant by ‘additive’. It is also unclear what was used for ‘expected mortality’ in the calculation of co-toxicity factor. I want to see explicit values and shouldn’t have to track down other literature to understand your study.

One important and potentially very interesting point I think you should raise is how these analyses and your results could differ considerably if you’d examined resistant Aedes aegypti. We might predict that how certain natural oils behave alone or in combination might change considerably if the mosquitoes being examined are resistant to pyrethrin. In particular, this paper should emphasize that future work should be done to elucidate synergistic the synergistic potential of these oils to potentially help overcome low levels of resistance. (For example, see Chansang et al. 2018, Parasites & Vectors.) You justify your study by stating that insecticide-resistant mosquitos are an increasing problem (e.g. line 27), but then don’t return to this point later on (and never mind the fact you didn’t actually look at resistant mosquitoes!)

Minor Points

Lines 16/17: “... in combination with a natural insecticide, natural pyrethrins, against the yellow...” The second ‘natural’ here seems redundant and could be removed?

Lines 21/22: I think “...synergized and antagonized...” would sound better as ‘synergized or antagonized”

Line 75: “potent synergism of various synthetic insecticides”, should maybe be “potent synergism with various synthetic insecticides” ?

Lines 92/93: “Oils were selected based on their novel chemistry and unique constituents compared to oils that were screened previously.” First, you should cite any papers here that you’re specifically considering [‘screened previously’]. Second, it’d be great to know what these ‘novel chemistry and unique constituents’ are! Perhaps a table/supplementary table would really be valuable to understand the motivation for selecting these specific oils.

I think if Figure 1 was divided into two separate components (an A & B), for 2 ug and 10 ug, respectively, it would make comprehension and interpretation easier.

Reviewer 2 Report

Dear authors,

This is a well written paper dealing with the screening of 20 essential oils in terms of their synergistic or antagonistic properties with natural pyrethrins against Aedes aegypti adults. The study was also included the evaluation of the co-toxicity factor method with respect to traditionally calculated synergism ratios. The findings are interesting since they show synergism or antagonistic effect of several plant derived oils with pyrethrins, in terms of knockdown and killing effect against mosquito adults, depending on the dose applied.

Further down, you will find some comments of minor importance and clarifications that may benefit the manuscript.

Line 77: Please change the number of reference Norris et al. to [6] from [10]

Line 126: It seems that the reference you are mentioning is [10], not [11]. Please check.

Line 142 (Results): You could limit the text in the results section. You don’t have to refer to so much data, since they are already presented in the tables. Please focus to the most critical ones.

Lines 310-312: It is stated that “Unexpectedly, some oils and PBO were synergistic only at the low dose, but not at the high dose, which may result from sequestration of pyrethrins at the cuticle”. Does this mean that excess oil or PBO containing pyrethrins did not penetrate the cuticle? Please clarify this assumption. This clarification may

Lines 343-345: It is stated that “It is possible that the slope values of combined mixtures may directly affect the degree of co-toxicity factors observed at low or high potency screening concentrations, in addition to any number of toxicokinetic/dynamic factors”. Could you please explain how the slope values that indicate how rapidly the response such as mortality is achieved, may affected the weak correlation between the co-toxicity factor method and the traditionally calculated synergism ratio at the high dose of 10 μg/insect?  

Reviewer 3 Report

The authors have conducted a thorough study to determine synergism or antagonism of various plant oils along with natural pyrethrins. The study provides good preliminary results which can be further explored for use of these oils as insecticides along with the traditional approach.

The authors have conducted the experiments and conducted the analysis with previously published methods (co-toxicity factor) and have included statements regarding the drawback for such analysis in conclusion.

Overall this is a good study which will provide a basis for future work in this area.

Reviewer 4 Report

Norris & Bloomquist present the results of laboratory bioassays in which 20 essential oils were tested for knockdown and mortality against non-fed female Ae. aegypti. The interaction of these oils was subsequently tested in mixtures with pyrethrins. They conclude that some oils could potentiate the insecticidal activity of the pyrethrins, with potential applications in vector control.
I was unable to fully review this study and I do not know if it could be suitable for publication as key details in the methodology and results were missing. The statistical analyses appear to be unsuitable and prone to type II errors.
I suggest that the manuscript be rejected. The authors could then include these details and new analyses and resubmit a carefully prepared manuscript.

Main points.
Experimental details are missing, including the number of replicates performed, the number of insects per replicate (cohort?), the controls performed (I'm assuming that ethanol controls were performed?). The control mortality is also missing from the Results section. This is important as the manual manipulation of adult mosquitoes could result in appreciable levels of mortality.
A large number of t-tests were performed to identify significant changes in knockdown and mortality in treated insects. Large numbers of independent pairwise comparisons markedly increase the risk of type II errors. The results should be subjected to ANOVA or GLM analyses as long as they meet the appropriate assumptions of normality and homoscedasticity, followed by a suitable multiple comparison procedure, e.g. Tukey. If desired, GLMs could be fitted using a binomial error structure.
The authors employ terminology that was unfamiliar. I suggest that they use the terminology in common usage in insecticide bioassays (see the excellent book Bioassays with Arthropods: 2nd Edition. Robertson, JL; Russell RM; Preisler HK; Savin NE (CRC Press, Boca Raton, FL. 2007, ISBN 978-0-8493-2331-7). Co-toxicity and synergism ratio should be replaced with commonly accepted terms (e.g., relative potency) where appropriate.
I have written numbered points on a scanned copy of the manuscript.

Numbered points (see scanned file)
1. Words that appear in the title should not appear in key words.
2. Essential oils are not a technology.
3. Did you use concentrations? You say that you used doses (LD25, LD50) in the text.
4. Please provide a citation for the mosquito rearing procedures.
5. Please use SI units for an international journal such as Insects (cm not inches).
6. What was the humidity range during the experiments? Was humidity controlled? I guess Florida has high humidity for most of the year?
7a. Line 116 says a volume of 0.2 µL was applied to each insect, so how was it possible to apply a DOSE of 2 and 10 µg/insect, this would be more than 2 and 10 µL assuming that the density of oil is less than that of water. Please clarify.
7b. How many insects in the cohorts?
8a. 16 fl oz = 470 mL capacity? Is a "deli pot" a plastic or paper tub or cup? Please provide information for readers outside the US.
8b. Please state how you applied such a tiny volume to each insect in a controlled and consistent manner.
9. How many replicates of each treatment? What did a replicate consist of?
10. What were your controls? Mosquitoes treated with ethanol alone?
11. What was the purpose of the cohort? Was this a group of insects to which all the treatments and controls were applied on a specific occasion? How many insects involved?
12. Please check my suggested rewriting of this sentence. Were insects incubated individually or in groups in cups/pots? How many per cup/pot?
13. I was unable to find the information on co-toxicity factors in the cited reference. Please check your citations.
14. Why? What is the statistical basis for considering values of 20 as indicators of synergism and values of less than -20 antagonism? This is an important point that needs clarification.
15. It was unclear to me whether LD25 and LD50 values were estimated in the present study or a previous study (not mentioned in Methods). Define OPTC - why was this correction performed?
16. Please state how you calculated "conventional" synergism ratios.
17. This appears to be an unsuitable application of large numbers of t-tests. I suggest you apply ANOVA or GLM models (as long as the data are suitable or can be transformed to meet the assumptions of ANOVA).
18. 2 or 10 µg/insect are DOSES not concentrations.
19. Table 1, what do ± values indicate? SD? SE?
20. What was the control exactly?
21. See point 18.
22. You used an acetone control? This is not mentioned in the Methods section, only ethanol.
23. What does statistically significant mortality mean?
24. Are you referring to Table 3 here?
25. I was unable to understand this text. Please clarify.
26. Table 3. Are you referring to Relative Potency here? These values appear to be ratios of LD50 values - see the book by Robertson et al., cited above.
27. State what the control was. Pyrethrins was a treatment, not a control per se.
28. Please show 95% CI for the "synergism ratio" (=relative potency?)
29. This text is long and highly repetitive as most of the information has already been mentioned on previous pages. Please remove or retain just the essential information.
30. Why present both r and R2 values? Please select one and eliminate the other.
31. Pyrethroids not tested in the present study.
32. Why are you interested in the use of synergists mixed with essential oils (?) as the use of essential oils as synergists with pyrethrins was the objective of the study (line 85). Please clarify or delete.
33. Please define EO.
34. Please explain what this substance is and what relevance it has to your study.
35. I did not understand this. Please reword.
36. I think the authors are over-reaching here. Such a demonstration would require additional studies on field efficacy, persistence and cost-benefit analyses.

Minor points
I have made numerous suggestions for improvements to the clarity and readability of the text (see scanned manuscript).
The references are not formatted for Insects (not scanned).
